# Chloride Ions, Vascular Function and Hypertension

**DOI:** 10.3390/biomedicines10092316

**Published:** 2022-09-18

**Authors:** Kenichi Goto, Takanari Kitazono

**Affiliations:** 1Department of Health Sciences, Graduate School of Medical Sciences, Kyushu University, Fukuoka 812-8582, Japan; 2Department of Medicine and Clinical Science, Graduate School of Medical Sciences, Kyushu University, Fukuoka 812-8582, Japan

**Keywords:** chloride, calcium-activated chloride channel, hypertension, Na^+^–K^+^–2Cl^−^ cotransporter 1, TMEM16A, smooth muscle

## Abstract

Blood pressure is determined by cardiac output and systemic vascular resistance, and mediators that induce vasoconstriction will increase systemic vascular resistance and thus elevate blood pressure. While peripheral vascular resistance reflects a complex interaction of multiple factors, vascular ion channels and transporters play important roles in the regulation of vascular tone by modulating the membrane potential of vascular cells. In vascular smooth muscle cells, chloride ions (Cl^−^) are a type of anions accumulated by anion exchangers and the anion–proton cotransporter system, and efflux of Cl^−^ through Cl^−^ channels depolarizes the membrane and thereby triggers vasoconstriction. Among these Cl^−^ regulatory pathways, emerging evidence suggests that upregulation of the Ca^2+^-activated Cl^−^ channel TMEM16A in the vasculature contributes to the increased vascular contractility and elevated blood pressure in hypertension. A robust accumulation of intracellular Cl^−^ in vascular smooth muscle cells through the increased activity of Na^+^–K^+^–2Cl^−^ cotransporter 1 (NKCC1) during hypertension has also been reported. Thus, the enhanced activity of both TMEM16A and NKCC1 could act additively and sequentially to increase vascular contractility and hence blood pressure in hypertension. In this review, we discuss recent findings regarding the role of Cl^−^ in the regulation of vascular tone and arterial blood pressure and its association with hypertension, with a particular focus on TMEM16A and NKCC1.

## 1. Introduction

Hypertension is the most prevalent and important risk factor for cardiovascular disease around the world [1], and cardiovascular complications associated with hypertension accounted for 8.5 million deaths worldwide in 2015 [2]. Nevertheless, global control (<140/90 mmHg) rates among subjects with hypertension in 2019 were only 23% for women and 18% for men [3], and thus more effective treatment strategies for hypertension control are urgently needed.

Lifestyle modifications are recommended for the treatment and prevention of hypertension and hypertension-associated cardiovascular diseases for all subjects, including subjects with high normal blood pressure and patients who are taking antihypertensive agents [4]. In particular, the restriction of dietary sodium chloride (NaCl) has been one of the major focus points among lifestyle modifications for the treatment and prevention of hypertension [5,6]. Indeed, numerous animal and human studies have established a causal relationship between dietary NaCl intake and hypertension as well as hypertension-associated cardiovascular diseases [7,8,9].

While it is generally assumed that sodium ions (Na^+^) but not chloride ions (Cl^−^) play a critical role in NaCl-induced hypertension [10,11], the copresence of Na^+^ and Cl^−^ has been reported to be requisite for the development or progression of hypertension in some animal models of hypertension, including desoxycorticosterone-induced hypertensive rats [12], Dahl salt-sensitive hypertensive rats [13,14] and stroke-prone spontaneously hypertensive rats [15]. Likewise, several studies have suggested the importance of Cl^−^ in NaCl-induced hypertension in humans [16,17,18]. These animal and human studies suggest that Na^+^ alone may not be sufficient, and that Cl^−^ may be indispensable or may act cooperatively with Na^+^ to give rise to NaCl-induced hypertension. A detailed description of the role of Cl^−^ in NaCl-induced hypertension in animals and humans can be found in an excellent review by McCallum et al. [19].

The precise mechanisms by which Cl^−^ contributes to the blood pressure rise in the above studies are yet to be determined, but the ability of Cl^−^ to modify vascular contractility may play a role. In vascular smooth muscle cells, the intracellular concentration of Cl^−^ is accumulated by anion exchangers and the anion–proton cotransporter system [20,21]. As the resting membrane potential of smooth muscle in vivo (e.g., −38 mV in the rat caudal artery [22]) is more negative than the reversal potential for Cl^−^ (e.g., −18 mV in the guinea pig vas deferens [23]), the opening of Cl^−^ channels leads to an efflux of Cl^−^ and depolarizes the membrane potential, which would then increase the open probability of L-type Ca^2+^ channels to trigger smooth muscle constriction [20,24].

Thus, in situations with increased intracellular Cl^−^ concentration or increased Cl^−^ channel activity in vascular smooth muscle cells, the driving force for the efflux of Cl^−^ is expected to increase, which in turn could facilitate membrane depolarization and vasoconstriction, and emerging evidence suggests that this scenario is indeed the case in some animal models of hypertension. In this review, we will discuss the possible involvement of Cl^−^ in the pathogenesis of hypertension. Particular emphasis is given to the roles of Ca^2+^-activated Cl^−^ channel transmembrane membrane 16A (TMEM16A; also known as Ano1) and Na^+^–K^+^–2Cl^−^ cotransporter 1 (NKCC1) in the increased vascular contractility during hypertension.

## 2. Role of Chloride Ions in Regulation of Vascular Tone and Blood Pressure

The vascular tone in vivo is regulated by perivascular nerves, including sympathetic, parasympathetic and non-adrenergic non-cholinergic nerves, and the corelease of norepinephrine and ATP from the sympathetic nerve terminals causes vascular smooth muscle membrane depolarization and subsequent constriction [25,26,27,28,29]. Although multiple ionic mechanisms would underpin the nerve-mediated vascular smooth muscle depolarization, several previous studies have suggested that nerve-mediated and exogenously applied norepinephrine-evoked smooth muscle depolarization could be at least partly due to the generation of Ca^2+^-activated Cl^−^ currents triggered by the Ca^2+^ release from the intracellular Ca^2+^ stores [24,30,31,32].

In addition to perivascular nerve-mediated regulation, myogenic response-mediated vascular smooth muscle depolarization and constriction in response to intravascular pressure change also contribute to the regulation of vascular tone [33]: in rat cerebral arteries, intravascular pressure-induced depolarization and constriction have been shown to be inhibited by two distinct Cl^−^ channel blockers, indanyloxyacetic acid (IAA-94) and 4,4’-diisothiocyanatostilbene-2,2’-disulphonic acid (DIDS), suggesting that the efflux of Cl^−^ ions through Cl^−^ channels could contribute to the myogenic response-mediated vasoconstriction [34]. Indeed, in support of this observation, efflux of Cl^−^ ions was associated with the myogenic constriction in the rat cerebral vascular bed [35]. Nevertheless, because subsequent studies performed in the rat cerebral arteries revealed that IAA-94 depresses L-type calcium current [36], and both IAA-94 and DIDS depress non-selective cationic current [37], the validity of the contribution of Cl^−^ currents to the myogenic response was called into question.

As such, despite a significant amount of physiological and pharmacological evidence showing that vascular Cl^−^ channels play a crucial role in regulating vascular tone, the absence of specific inhibitors and the lack of the molecular identities of the channels make it difficult to reach indisputable conclusions. Among other things, there has been a debate regarding the molecular identity of CaCCs ever since the initial report by Byrne and Large in 1987 [38]. Indeed, several proteins have been proposed as the molecular counterpart of CaCCs, and these include CLCA, CLC-3, TWEENTY and bestrophins [39]. However, three independent groups revealed in 2008 that the TMEM16A protein is a molecular counterpart for CaCCs [40,41,42].

Since these 2008 reports, many studies have confirmed that TMEM16A generates functional CaCC currents in a number of vascular smooth muscle cells and thereby regulates agonist-induced vasoconstriction [21,43,44,45]. Moreover, it has been revealed that TMEM16A also contributes to intravascular pressure-induced myogenic depolarization and vasoconstriction in the cerebral arteries and renal arterioles of rats [46,47]. Thus, it appears likely that the TMEM16A in vascular smooth muscle cells plays a critical role in regulating vascular tone and blood pressure. Support for this notion comes from the fact that conditional knockout mice of TMEM16A in vascular smooth muscle cells shows a complete deficiency of CaCC currents, decreased responsiveness to vasoconstrictor stimuli and reduced systemic blood pressure [48].

## 3. Alterations in Vascular Chloride Channels and Transporters in Hypertension

### 3.1. Ca^2+^-Activated Chloride Channels (CaCCs) in Vascular Smooth Muscle Cells

It is generally accepted that essential hypertension is characterized by an increased peripheral resistance [49,50]. The increased peripheral resistance in hypertension is determined by an integral and complex interplay between various pathogenic factors, including increased sympathetic nervous activity, enhanced calcium ion mobilization in vascular smooth muscle cells, increased calcium sensitivity of vascular smooth muscle cells and reduced production of endothelium-derived relaxing factors, to name a few [50,51]. Among these factors, alterations in the function of vascular ion channels during hypertension contribute to the increased peripheral resistance by shifting the membrane potential to depolarized levels [22,50,52].

While many studies have demonstrated downregulation of the expression and/or function of vascular potassium (K^+^) channels in hypertension [50,51,53,54], emerging evidence reveals an upregulation of expression and/or function of CaCCs in vascular smooth muscle cells of spontaneous hypertensive rats (SHRs), a genetic model of human essential hypertension. Although a previous study suggested an increased activity of CaCCs in vascular smooth muscle cells of SHRs [55], the molecular identity of the CaCCs observed in that study was unclear at the time. A subsequent study by Wang et al. for the first time revealed that TMEM16A is the molecular counterpart for the increased activity of CaCCs in vascular smooth muscle cells of SHRs, and that TMEM16A protein expression is significantly upregulated in the aorta, the carotid arteries, the hindlimb arteries and the mesenteric arteries of SHRs compared to those of normotensive Wistar Kyoto (WKY) rats [56] (Table 1). Consistent with the seminal findings of Wang and colleagues [56], the increased TMEM16A expression levels and the resultant potentiation of vasoconstrictions have also been reported in smooth muscle cells of the coronary arteries [57] and the renal arterioles [47] of SHRs (Table 1).

Importantly, the increased expression and function of TMEM16A appear to be associated with blood pressure elevation in SHRs: the in vivo knockdown of TMEM16A by small interfering RNA (siRNA) transfection prevented blood pressure rise, and the in vivo inhibition of TMEM16A activity by T16A_inh_-A01, a TMEM16A inhibitor, reduced blood pressure in SHRs [56] (Table 1). Similarly, a recent study in SHRs showed that in vitro treatment of mesenteric resistance arteries with TM_inh_-23, a small molecule inhibitor of vascular smooth muscle TMEM16A, blocked vascular smooth muscle constriction in response to vasoconstrictor stimuli, and in vivo treatment with TM_inh_-23 reduced blood pressure in SHRs with minimal blood pressure change in normotensive rats and mice [58] (Table 1). Although the greater blood pressure lowering effect of TM_inh_-23 in SHRs appears to be due to an increased sensitivity of TMEM16A to TM_inh_-23 [58], the mechanisms underlying the increased sensitivity of TMEM16A are unclear and warrant further investigations. Together, these findings implicate vascular smooth muscle CaCC TMEM16A as a possible contributor in the pathogenesis of hypertension in SHRs.

In rat basilar arteries of 2-kidney, 2-clip (2K2C) renal hypertensive rats, exogenously applied angiotensin II (Ang II) induced vasoconstriction that was sensitive to T16A_inh_-A01, and Ang II evoked TMEM16A-mediated CaCC currents in rat basilar smooth muscle cells [59]. These findings suggest that CaCC TMEM16A modulates the vasocontractility of basilar arteries of 2K2C renal hypertensive rats; however, in sharp contrast with SHRs, the activity of CaCCs was decreased gradually during the development of hypertension, and the CaCCs’ current density was negatively correlated with blood pressure levels, in basilar arteries of 2K2C renal hypertensive rats [60] (Table 1). Moreover, the TMEM16A protein expression in the smooth muscle layer of the basilar artery decreased during the development of hypertension in 2K2C renal hypertensive rats [59,60] (Table 1).

It is not clear why the activity and the expression of CaCC TMEM16A changed in the opposite direction between SHRs and 2K2C renal hypertensive rats, but the difference might be explained by the different levels of activity of the renin–angiotensin system (RAS) in the vasculature: while the plasma and tissue RASs are suppressed in SHRs [61], the RAS components—particularly the vascular Ang II concentration—are increased in 2K2C renal hypertensive rats [62]. As Ang II decreased TMEM16A expression in some vascular smooth muscle cells, including those from rat basilar arteries [59,60,63], an increase in vascular Ang II concentration in the basilar arteries of 2K2C renal hypertensive rats might downregulate TMEM16A expression and hence reduce the CaCCs’ current in this model.

It has been reported that the perivascular sympathetic nerves exert an abnormal trophic influence on the vascular smooth muscle membrane properties of SHRs [64], and a recent report showed that the expression and contractile function of the CaCC TMEM16A in rat arteries were reduced due to the trophic influence of sympathetic nerves during postnatal maturation [65]. Therefore, we speculate that the expression and function of CaCC TMEM16A might also be decreased along with the longer duration of hypertension in SHRs because of the persistent abnormal trophic influence of the sympathetic nerves. This hypothesis might be supported by the observation that the contribution of CaCCs to norepinephrine-induced vasoconstriction in the femoral arteries was decreased in 12-month-old SHRs compared to that of 6-month-old SHRs [66].

TMEM16A may modulate vascular contractility in cooperation with other ion channels in certain vascular beds. Thus, in rat mesenteric and tail arteries, TMEM16A modulates vascular contractility, at least in part, by positively regulating the expression and function of vascular L-type Ca^2+^ channels [67,68]. In another study in rat cerebral arterial smooth muscle cells, transient receptor potential canonical 6 channel (TRPC6) and TMEM16A were found to be spatially localized, and TRPC6 activation led to a local elevation of Ca^2+^, which in turn activated nearby TMEM16A, leading to vasoconstriction [69]. As the function and expression of both L-type Ca^2+^ channels [70,71] and TRPC6 [72] have been reported to be upregulated in hypertensive rats, it is intriguing to speculate that these mutual interactions of TMEM16A with other vascular ion channels function cooperatively to augment vasoconstriction and hence increase blood pressure in hypertension.

It has been reported that phosphatidylinositol 4,5-bisphosphate (PIP_2_), a phospholipid of the plasma membrane, regulates ion channel activity in various cell types [73], and several studies reported that PIP_2_ acts as a positive modifier of TMEM16A [74,75,76]. By contrast, the TMEM16A-mediated CaCC current was not augmented, but rather inhibited by PIP_2_ in rat pulmonary artery smooth muscle cells [77]. The reason for the discrepancy is not clear. Nevertheless, a previous report suggests that a significant difference exists between WKY and SHR aortas regarding the PIP_2_ hydrolysis response following stimulation with norepinephrine [78], indicating the need for further research to understand the possible regulation of TMEM16A by PIP_2_ in blood vessels in hypertension.

Recent evidence suggests that inositol 1,4,5-trisphosphate receptors (IP_3_Rs) are spatially colocalized with TMEM16A proteins in nociceptive sensory neurons [79]. If the same holds true in vascular smooth muscle cells, IP_3_-induced Ca^2+^ release from intracellular Ca^2+^ stores would activate nearby TMEM16A, and alterations in this signaling pathway might contribute to the TMEM16A-mediated vasoconstriction in SHRs. Indeed, it has been reported that IP_3_R channels are upregulated in vascular smooth muscle in hypertension, resulting in enhanced IP_3_-induced Ca^2+^ release and increased vasoconstriction [80].

To sum up, while there is a growing body of evidence that CaCC TMEM16A contributes to the increased vascular contractility and elevated blood pressure in SHRs, it is currently unclear whether the upregulation of TMEM16A is specific to SHRs or is present in other hypertensive animal models, and further studies will be needed to clarify the molecular mechanisms that regulate TMEM16A activity during hypertension.

### 3.2. Ca^2+^-Activated Chloride Channels (CaCCs) in Vascular Endothelial Cells

In addition to their expression in vascular smooth muscle cells, CaCCs have been reported to be present in some vascular endothelial cells [81,82,83,84]. Although the physiological role of endothelial CaCCs is still not well understood, the endothelial CaCCs may contribute to the regulation of the resting membrane potential of the endothelial cells. Indeed, in mouse brain capillary endothelial cells, pharmacological blockade or knockdown of TMEM16A with siRNA induced membrane hyperpolarization, suggesting that the activation of endothelial CaCCs acts to depolarize the membrane potential of the endothelial cells [83]. Further support for this notion comes from the study by Yamamoto et al. [85]. They found that, in the isolated endothelium of guinea pig mesenteric arteries, ACh increased the intracellular concentration of Ca^2+^, which subsequently activated endothelial small conductance Ca^2+^-activated K^+^ channels (SK_Ca_s), intermediate conductance K_Ca_ (IK_Ca_) and CaCC simultaneously, and the endothelium-dependent hyperpolarization (EDH) through the activation of both SK_Ca_ and IK_Ca_ was counteracted by the opposing membrane depolarization evoked by the activation of CaCCs [85].

With respect to the alteration of endothelial CaCCs in hypertension, we have previously shown a functional upregulation of endothelial CaCCs in mesenteric resistance arteries of SHRs [86]. In that study, after blockade of EDH with K_Ca_ channel inhibitors, iontophoresed acetylcholine (ACh) evoked a rapid and substantial membrane depolarization in mesenteric resistance arteries of SHRs, but only negligible slow depolarization was detected in those of WKY rats [86,87] (Figure 1).

As the estimated reversal potential of the ACh-evoked depolarization in that study was −18 mV [86], which agrees closely with that reported for Cl^−^ ions [23,88], and the ACh-evoked depolarization was abolished by endothelium denudation, or reduced either by replacement of external Cl^−^ ions with impermeant anions or by treatment with the CaCC inhibitors niflumic acid or flufenamic acid, the ACh-evoked depolarization appears to be, at least in part, generated through the activation of endothelial CaCCs in the mesenteric resistance arteries of SHRs [86]. Moreover, the inhibition of the ACh-evoked depolarization by CaCC inhibitors improved the impaired ACh-induced EDH in mesenteric arteries of SHRs, suggesting that an increased activity of endothelial CaCCs may be responsible for the impairment of EDH (Figure 2) (Table 2).

As endothelial cells and adjacent smooth muscle cells are electrically coupled via myoendothelial gap junctions in rat mesenteric arteries [89,90,91], the impaired EDH leads to attenuated EDH-mediated relaxation and hence to endothelial dysfunction in SHRs [86]. Although some studies have reported that the inhibition of volume-activated Cl^−^ channels potentiates K^+^-induced, EDH-mediated relaxation in rat mesenteric arteries [92,93], the involvement of the volume-activated Cl^−^ channels in the ACh-evoked depolarization in mesenteric arteries of SHRs is not likely because the volume-regulated Cl^−^ channel inhibitor NPPB had no effect on the ACh-evoked depolarization in this vascular bed [86].

A negative causal link between the activity of endothelial CaCCs, specifically TMEM16A, and endothelial function has also been reported in other studies [82,84]. Thus, in Ang II-induced hypertensive mice, in which the expression of vascular endothelial TMEM16A is increased, the endothelial-specific TMEM16A knockout ameliorated endothelial function and lowered the systolic blood pressure, whereas the endothelial-specific TMEM16A overexpression deteriorated endothelial function and further elevated the systolic blood pressure [82], and these interactions appear to be related to the facilitating effects of TMEM16A on reactive oxygen species generation via Nox2-containing NADPH oxidase [82] (Table 2).

Another study showed that overexpression of TMEM16A in human pulmonary endothelial cells led to a decrease in ACh-induced NO production [84]. Taken together, these findings suggest that upregulation of endothelial CaCC TMEM16A may contribute to the impaired endothelial function, and if so, that it likely does so via a reduction in the activity of EDH and/or NO; finally, the results suggest that such a reduction in EDH and/or NO activity may be at least partly responsible for the elevated blood pressure in hypertension (Figure 3).

### 3.3. Na^+^–K^+^–2Cl^−^ Cotransporter1 (NKCC1)

NKCC1 located on vascular smooth muscle cells functions to accumulate intracellular Cl^−^ [20,21]. The most compelling evidence of the functional role of NKCC1 in the regulation of vascular tone and arterial blood pressure comes from studies on NKCC1 knockout mice: the systolic blood pressure was significantly reduced in NKCC1 knockout mice compared to wild-type mice [94], and treatment with bumetanide, an inhibitor of NKCC1 [95], inhibited the vascular contractile activity and lowered mean arterial blood pressure in wild-type mice, with the effects being lost in NKCC1 knockout mice [94,96]. Thus, theoretically, an increase in the activity of the vascular smooth muscle NKCC1 could augment vascular contractility and subsequently lead to enhanced blood pressure, and this is indeed the case in several types of hypertensive rats.

In some experimental models of hypertensive rats, including SHRs [97,98,99], Milan hypertensive rats [100] and deoxycorticosterone acetate (DOCA) salt hypertensive rats [101], increase in the activity of NKCC in vascular smooth muscle cells has been reported (Table 3). Interestingly, Lee et al. reported that the mRNA and protein expression levels of NKCC1 were epigenetically upregulated in the aorta of SHRs due to *Nkcc1* gene promoter hypomethylation [102], and the *Nkcc1* gene promoter hypomethylation resulted from the decreased activity of DNA methyltransferase 3B [103] (Table 3). Likewise, an epigenetic upregulation of NKCC1 via histone modifications was reported in the aorta of Ang II-induced hypertensive rats [104] (Table 3). 

In addition to the epigenetic upregulation of NKCC1, another factor may also contribute to the increase in the activity of NKCC1 during hypertension. In fact, some studies have suggested the possible positive regulation of vascular NKCC1 by with-no-lysine kinase (WNK) and sterile-20-related praline–alanine-rich kinase (SPAK): heterozygous WNK1 knockout mouse aorta exhibited reduced phosphorylation of downstream SPAK and NKCC1, leading to decreased responses to vasoconstrictive stimuli [105]. Similarly, the aorta of SPAK knockout mice exhibited reduced phosphorylation of NKCC1 and decreased NKCC1-mediated vascular constriction, and the SPAK knockout mice had low blood pressure [106]. Moreover, activation of the WNK3/SPAK/NKCC1 pathway has been shown to be involved in both the Ang II-induced aortic constriction and Ang II-induced blood pressure rise in mice [107].

These observations suggest that the WNK/SPAK signaling pathway positively regulates the vascular NKCC1 toward vasoconstriction and hypertension. Interestingly, mutations of WNK have been found in patients with familial hyperkalemic hypertension, a form of monogenic hypertension [108]. Nevertheless, there is no evidence to date that demonstrates changes in the WNK/SPAK pathway in animal models of polygenic hypertension such as SHRs or in human essential hypertension.

The studies mentioned above have demonstrated that the expression and/or the function of NKCC1 are upregulated in vascular smooth muscle cells of hypertensive rats. Then, the question arises how the upregulation of NKCC in hypertension contributes to the augmented vascular contractility and increased blood pressure. It has been reported that the intracellular concentration of Cl^−^ is increased in vascular smooth muscle cells of DOCA salt hypertensive rats because of the increase in the activity of NKCC [101].

The increase in the intracellular Cl^−^ concentration would increase the driving force for Cl^−^ efflux via Cl^−^ channels such as CaCCs upon vasoconstrictor stimulation, and the increase in Cl^−^ efflux would make the membrane potential more depolarized [20], which in turn would enhance the open probability of voltage-gated L-type Ca^2+^ channels, leading to an increase in vascular tone. In support of this notion, we have shown that the inhibition of the NKCC with bumetanide, an inhibitor of NKCC1 [95], significantly reduced the CaCC-mediated membrane depolarization and constriction in vascular smooth muscle of SHRs [86]. Since, as discussed in the previous section, CaCCs are also functionally upregulated in the vasculature of hypertensive rats, we propose that the enhanced activities of NKCC1 and CaCCs act additively and sequentially to increase vascular contractility and hence blood pressure in hypertension (Figure 4).

## 4. Clinical Perspectives

While many animal studies suggest that the upregulation of TMEM16A and NKCC1 could contribute to the increased vascular contractility and elevated blood pressure in hypertension as mentioned in the preceding sections, there is very little information concerning their possible involvement in the pathogenesis of hypertension in humans. Interestingly, however, two independent population-based studies reported that some genetic variants of TMEM16A were significantly associated with hypertension in humans [57,109]. Further exploration of the functional impact of the SNP in the TMEM16A coding region could provide a clue to understand the pathophysiological role of TMEM16A in human hypertension.

In addition, it has been reported that there was a positive association between hyperchloremia and in-hospital mortality in hospitalized patients [110]. Moreover, in a recent study in patients with chronic kidney disease, hyperchloremia was an independent predictor of hypertension and proteinuria [111]. Taking these observational studies together, it might be possible to speculate that hyperchloremia might lead to blood pressure elevation and hence to poor prognosis. By contrast, in outpatients with hypertension [112] or chronic heart failure [113], hypochloremia was a predictor of mortality [112,113] but the level of Cl^−^ was not associated with the level of blood pressure [112]. Thus, while these findings indicate that serum Cl^−^ alterations are associated with poor prognosis in patients with elevated cardiovascular risk, the ability of changes in serum Cl^−^ concentration to affect blood pressure is not clear. Further studies are needed to clarify the role of serum Cl^−^ concentrations on blood pressure regulation and its association with long-term prognosis in patients with elevated cardiovascular risk.

## 5. Conclusions

Accumulating experimental evidence suggests that Cl^−^ plays an important role in the regulation of vascular tone through its ability to depolarize vascular smooth muscle cells, and the increased contribution of Cl^−^ to arterial constriction appears to be associated with the development and progression of hypertension. Of note, there is a growing body of evidence that the upregulation of CaCC TMEM16A in the vasculature contributes to the increased vascular contractility and elevated blood pressure in genetically hypertensive rats. In addition, the increased activity of NKCC1 may also promote hypertension as the result of a robust accumulation of intracellular Cl^−^ in vascular cells.

Nevertheless, much remains to be determined about the precise molecular mechanisms underlying the increased activity of TMEM16A and NKCC1 as well as their interactions with other signaling pathways during hypertension, and most importantly the pathophysiological roles of these molecules in human hypertension. Further exploration of the arterial tone regulation by Cl^−^ may facilitate a better understanding of the pathogenesis of hypertension, which may help to develop a novel therapeutic strategy to tackle hypertension and hypertension-associated cardiovascular diseases.

## Figures and Tables

**Figure 1 biomedicines-10-02316-f001:**
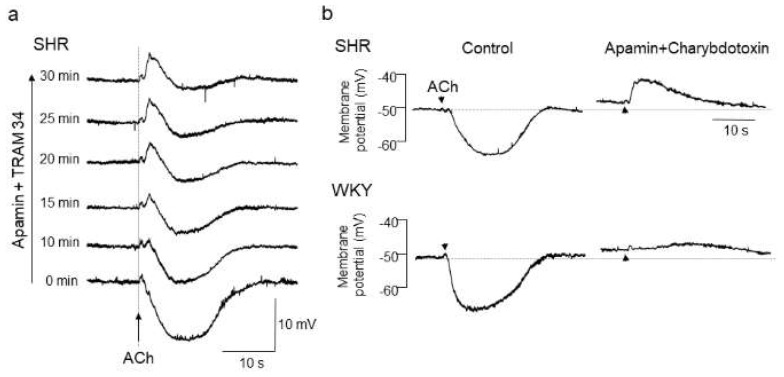
Acetylcholine (ACh)-evoked depolarization in mesenteric arteries of spontaneously hypertensive rats (SHRs). (**a**) A hidden depolarization emerged after blockade of endothelium-dependent hyperpolarization (EDH) with apamin (0.5 μmol/L, a small-conductance Ca^2+^-sensitive K^+^ channel (K_Ca_) inhibitor) plus TRAM34 (100 nmol/L, an intermediate-conductance K_Ca_ inhibitor) in mesenteric arteries of SHRs. All recordings were from the same cell. (**b**) ACh-evoked depolarization in the presence of apamin (0.5 μmol/L) plus charybdotoxin (60 nmol/L, a large and intermediate-conductance K_Ca_ inhibitor) was larger in amplitude and faster in time course in SHRs than in Wistar Kyoto (WKY) rats. Each paired recording was from the same preparation. Indomethacin (10 μmol/L) and N^ω^-nitro-L-arginine methyl ester (100 μmol/L) were present throughout. Arrows, application of ACh. Modified from Goto et al. [87].

**Figure 2 biomedicines-10-02316-f002:**
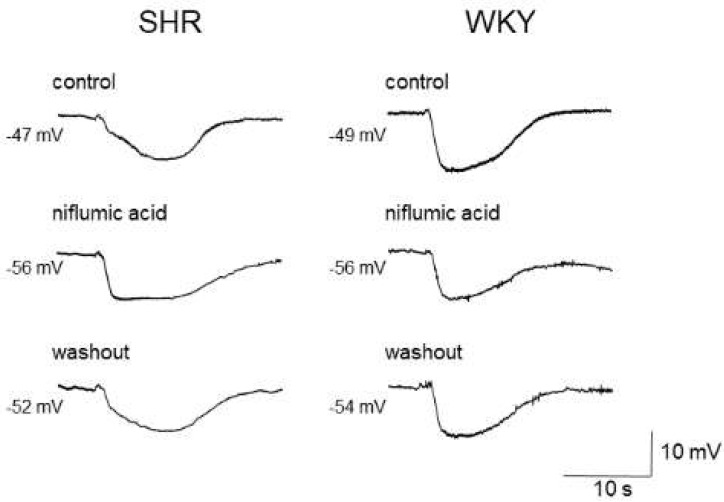
Effects of niflumic acid on acetylcholine-induced, endothelium-dependent hyperpolarization (EDH) in mesenteric arteries of spontaneously hypertensive rats (SHRs) and Wistar Kyoto (WKY) rats. Niflumic acid (50 μmol/L) improved EDH in SHRs but not in WKY rats. Each paired recording was from the same preparation. Indomethacin (10 μmol/L) and N^ω^-nitro-L-arginine methyl ester (100 μmol/L) were present throughout.

**Figure 3 biomedicines-10-02316-f003:**
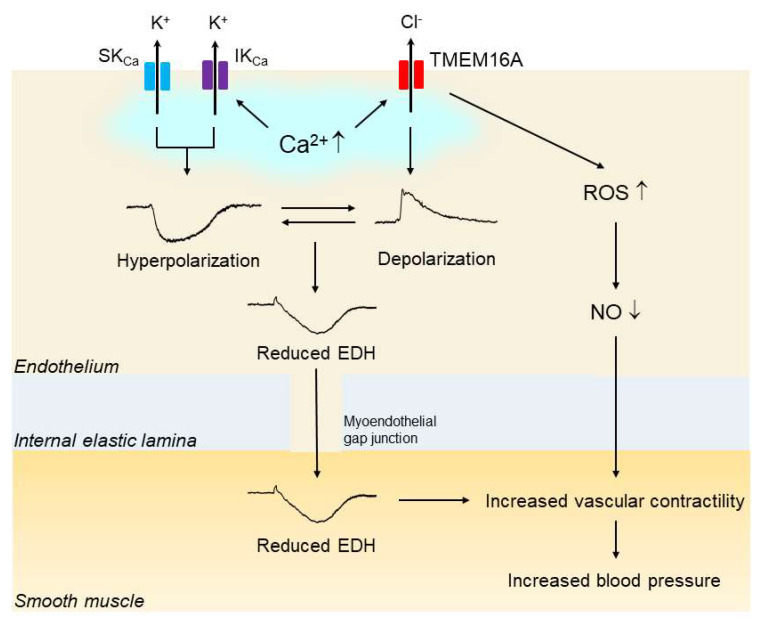
Upregulation of endothelial TMEM16A impairs endothelial function in hypertension. In hypertension, the expression and function of vascular endothelial Ca^2+^-activated Cl^—^ channel TMEM16A are increased. Endothelial stimulation with agonists and shear stress increases the intracellular Ca^2+^concentration, which subsequently activates endothelial small-conductance Ca^2+^-activated K^+^ channels (SK_Ca_s), intermediate-conductance K_Ca_ (IK_Ca_) and TMEM16A simultaneously. The endothelium-dependent hyperpolarization (EDH) through the activation of both SK_Ca_ and IK_Ca_ is reduced by the opposing membrane depolarization evoked by the activation of TMEM16A. In addition, activation of TMEM16A may facilitate the generation of reactive oxygen species (ROS) through Nox2-containing NADPH oxidase, leading to reduced bioavailability of nitic oxide (NO). Impaired EDH and/or NO could be at least partly responsible for the blood pressure rise in hypertension.

**Figure 4 biomedicines-10-02316-f004:**
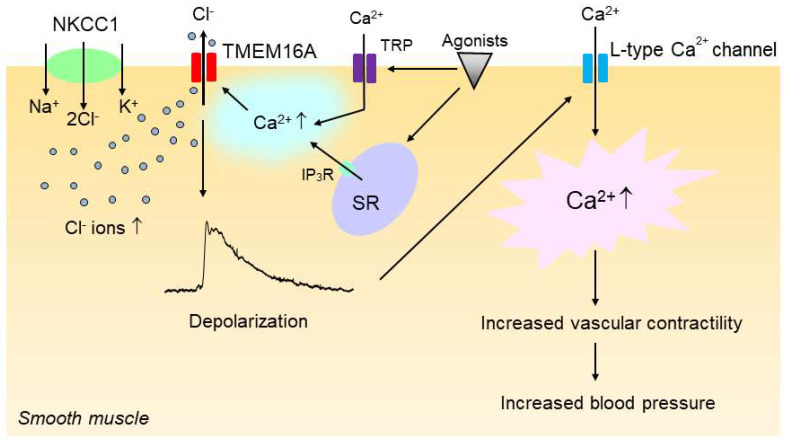
Upregulation of smooth muscle Na^+^–K^+^–2Cl^−^ cotransporter 1 (NKCC1) and TMEM16A additively and sequentially increases vascular contractility in hypertension. In hypertension, the intracellular concentration of Cl^−^ is increased in vascular smooth muscle cells because of the increased activity of NKCC1. The increase in the intracellular Cl^−^ concentration then increases the driving force for Cl^−^ efflux via the Ca^2+^-activated Cl^−^ channel TMEM16A when TMEM16A is activated by intracellular Ca^2+^ rise upon stimulation with vasoconstricting agonists, which in turn induces membrane depolarization. TMEM16A might be regulated by a local Ca^2+^ increase that could be generated by IP_3_R channels on the sarcoplasmic reticulum (SR) and/or transient receptor potential (TRP) channels. The membrane depolarization would then enhance the open probability of voltage-gated L-type Ca^2+^ channels, leading to an increase in vascular contractility and blood pressure.

**Table 1 biomedicines-10-02316-t001:** Alterations in vascular smooth muscle Ca^2+^-activated Cl^−^ channels during hypertension.

Animals	Alterations in Vascular Smooth Muscle CaCCs during Hypertension	Ref.
SHRs	Increased TMEM16A expression and function in aorta, carotid arteries, hindlimb arteries and mesenteric arteries	[56]
	Increased TMEM16A expression and function in coronary arteries	[57]
	Increased TMEM16A expression and function in renal arterioles	[47]
	Knockdown of TMEM16A by siRNA transfection lowered blood pressure	[56]
	Inhibition of TMEM16A activity by T16A_inh_-A01 lowered blood pressure	[56]
	Treatment of mesenteric resistance arteries with TM_inh_-23 blocked vasoconstriction	[58]
	Inhibition of TMEM16A activity by TM_inh_-23 lowered blood pressure	[58]
2K2C renal hypertensive rats	Reduced TMEM16A expression and function in basilar arteries during the development of hypertension	[59,60]

CaCCs, Ca^2+^-activated Cl^−^ channels; SHRs, spontaneously hypertensive rats; 2K2C, 2-kidney, 2-clip.

**Table 2 biomedicines-10-02316-t002:** Alterations in vascular endothelial Ca^2+^-activated Cl^−^ channels during hypertension.

Animals	Alterations in Endothelial CaCCs during Hypertension	Ref.
SHRs	Increased CaCC function in endothelium of mesenteric arteries	[86]
	Increased CaCC function, reduced EDH in mesenteric arteries	[86]
Ang Ⅱ-induced hypertensive mice	Increased TMEM16A expression in endothelium of aorta	[82]
	Endothelial-specific TMEM16A knockout ameliorated endothelial function and lowered blood pressure	[82]
	Endothelial-specific TMEM16A overexpression deteriorated endothelial function and elevated blood pressure	[82]

CaCCs, Ca^2+^-activated Cl^−^channels; SHRs, spontaneously hypertensive rats; EDH, endothelium-dependent hyperpolarization; Ang Ⅱ, angiotensin Ⅱ.

**Table 3 biomedicines-10-02316-t003:** Alterations in vascular smooth muscle NKCC1 during hypertension.

Animals	Alterations in Vascular Smooth Muscle NKCC1 during Hypertension	Ref.
SHRs	Increased NKCC1 function in aorta and carotid arteries	[97,98,99]
	Epigenetic upregulation of aorta NKCC1 due to *Nkcc1* gene promoter hypomethylation	[102]
	*Nkcc1* gene promoter hypomethylation resulted from the decreased activity of DNA methyltransferase 3B	[103]
Milan hypertensive rats	Increased NKCC1 function in thoracic aorta	[100]
DOCA salt hypertensive rats	Increased NKCC1 function in saphenous branch of femoral arteries	[101]
Ang Ⅱ-induced hypertensive rats	Epigenetic upregulation of aorta NKCC1 due to histone modifications	[104]

NKCC1, Na^+^–K^+^–2Cl^−^ cotransporter 1; SHRs, spontaneously hypertensive rats; DOCA, deoxycorticosterone acetate; Ang Ⅱ, angiotensin Ⅱ.

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
