# Peer review of "Chloride Ions, Vascular Function and Hypertension"

_biomedicines, 2022, doi:10.3390/biomedicines10092316_

Round 1

Reviewer 1 Report

The review of Goto and Kitazono entitled:” Chloride Ions, Vascular Function and Hypertension”  draws attention to the Ca2+-activated Cl channel (TMEM16A) and Na+- K+ - 2Cl contransporter1 (NKCC1) in the context of vascular contractility and hence blood pressure regulation. The authors appropriately discussed the role of Cl- in the regulation of vascular tone and arterial blood pressure and its association with hypertension. The list of human diseases known to be associated with defects in ion channels has grown considerably during the past years. Therefore, these proteins represent promising targets for the development of novel therapeutic agents for the near future. Reading the manuscript and appreciating the arguments, I identified the following minor issues which need to be addressed by the authors. In summary, the manuscript is promising and authors should be encouraged to revise it.

Minor concerns:

1. Line 124-131: TMinh-23 reduced the blood pressure in SHR but had a minimal effect in normotensive rats and mice. When TMinh-23 inhibits the TMEM16A protein, I am curious why its effect was minimal in normotensive rats and mice. In both SHR and normotensive rats, TMEM16A is involved in vascular tone regulation. How could be this phenomenon explained? Is it anything known about the sensitivity of the TMEM16A protein from SHR and normotensive rats to TMinh-23? Is it feasible to assume that such sensitivity of TMEM16A in SHR is increased by phosphorylation, for example?

2. Line 171: In respect to the possible regulation of TMEM16A by PIP2 in blood vessels, I would like to draw attention to the possible role of IP3R channels. There are intracellular Ca2+ channels, and are activated by IP3 produced by PIP2 hydrolysis. When IP3R channels become activated, a local Ca2+ release mediated by these channels might activate TMEM16A proteins. Is it anything known about the spatial colocalization of IP3R channels and TMEM16A proteins?

3. Figure 4: It is not clear what is the source of Ca2+ near the TMEM16A protein and whether it is related to the L-type Ca2+ channel or the ryanodine receptor (RYR), which is activated by Ca2+ entering the cell thought the L-type Ca2+ channel. It seems to me that TMEM16A might be regulated by a local Ca2+ increase that could be generated by TRP channels (as was mentioned in the manuscript) or potentially IP3R channels. The Ca2+ current via the L-type Ca2+ channels is likely not involved in this regulatory pathway.  This issue should be addressed in the Figure 4.

Author Response

Response to Reviewers

We wish to express our appreciation to the reviewers and the editor to improve the presentation and understanding of our data. We believe that the comments and suggestions were valuable and we feel the comments have helped us significantly improve the paper. We have considered all of the points raised by the reviewers and provide point-by-point replies to the issues and points the reviewers addressed. As per request of a reviewer, Tables (Table 1, 2 and 3) have been added in the revised version of our manuscript. In addition, a new section titled ‘Clinical perspective’ has been included.

# Reviewer 1

Thank you for the reviewer’s positive comments. We highly appreciate the constructive and pertinent comments of the reviewer. We have attempted to answer each of the comments raised. We feel the comments have helped us significantly improve the paper.

Comment 1:

Line 124-131: TMinh-23 reduced the blood pressure in SHR but had a minimal effect in normotensive rats and mice. When TMinh-23 inhibits the TMEM16A protein, I am curious why its effect was minimal in normotensive rats and mice. In both SHR and normotensive rats, TMEM16A is involved in vascular tone regulation. How could be this phenomenon explained? Is it anything known about the sensitivity of the TMEM16A protein from SHR and normotensive rats to TMinh-23? Is it feasible to assume that such sensitivity of TMEM16A in SHR is increased by phosphorylation, for example?

Response 1:

As the reviewer has correctly pointed out, the sensitivity of the TMEM16A protein was more sensitive to TMinh-23 in SHR compared with normotensive rats (ref. 58, Figure 7c), and this might be due to the increased phosphorylation of TMEM16A protein in SHR. Unfortunately, however, we could not find any literature evidence to support this notion nor any alternative explanation. We have briefly added this point in the revised manuscript as follows:

Lines 146-149: Although the greater blood pressure lowering effect of TMinh-23 in SHR appears to be due to an increased sensitivity of TMEM16A to TMinh-23 [58], the mechanisms underlying the increased sensitivity of TMEM16A are unclear and warrant further investigations.  

Comment 2:

Line 171: In respect to the possible regulation of TMEM16A by PIP2 in blood vessels, I would like to draw attention to the possible role of IP3R channels. There are intracellular Ca2+ channels, and are activated by IP3 produced by PIP2 hydrolysis. When IP3R channels become activated, a local Ca2+ release mediated by these channels might activate TMEM16A proteins. Is it anything known about the spatial colocalization of IP3R channels and TMEM16A proteins?

Response 2:

We appreciate the pertinent comment of the reviewer. Yes, the spatial colocalization of IP3R channels and TMEM16A proteins has been reported in nociceptive sensory neurons (Jin X et al., Sci Signal. 2013;6:ra73). If same holds true in vascular smooth muscle cells, IP3-induced Ca2+ release from intracellular Ca2+ store would activate nearby TMEM16A, and alterations in this signaling pathway might contribute to the functional upregulation of TMEM16A in SHR. Indeed, it has been reported that IP3R channels are up-regulated in vascular smooth muscle in hypertension, resulting in enhanced IP3-induced Ca2+ release and increased vasoconstriction (Abou-Saleh H et al., J Biol Chem. 2013;288:32941-51). We have added the following sentences in the text as follows:  

Lines 190-195: Recent evidence suggests that inositol 1,4,5‐trisphosphate receptors (IP3R) are spatially colocalized with TMEM16A proteins in nociceptive sensory neurons [79]. If same holds true in vascular smooth muscle cells, IP3-induced Ca2+ release from intracellular Ca2+ store would activate nearby TMEM16A, and alterations in this signaling pathway might contribute to the TMEM16A-mediated vasoconstriction in SHR. Indeed, it has been reported that IP3R channels are up-regulated in vascular smooth muscle in hypertension, resulting in enhanced IP3-induced Ca2+ release and increased vasoconstriction [80].    

Comment 3:

Figure 4: It is not clear what is the source of Ca2+ near the TMEM16A protein and whether it is related to the L-type Ca2+ channel or the ryanodine receptor (RYR), which is activated by Ca2+ entering the cell thought the L-type Ca2+ channel. It seems to me that TMEM16A might be regulated by a local Ca2+ increase that could be generated by TRP channels (as was mentioned in the manuscript) or potentially IP3R channels. The Ca2+ current via the L-type Ca2+ channels is likely not involved in this regulatory pathway.  This issue should be addressed in the Figure. 4.

Response:

We appreciate the appropriate comments of the reviewer. We agree with the reviewer’s comments that TMEM16A might be regulated by a local Ca2+ increase that could be generated by IP3R channels and/or TRP channels. As per suggested, the mechanism regulating TMEM16A channels have now been included in the Figure 4 and the legend of Figure 4.

Reviewer 2 Report

To:

Editorial Board

Biomedicines

Title: “Chloride Ions, Vascular Function and Hypertension”

Dear Editor,

I read this paper and I think that:

-       I think that authors should include clinical studies. The inclusion of population studies would increase the appeal of the text and the scientific background.

-       Authors should include tables which gather the main findings from literature. This would increase the readability of the text.

-       The authors should better discuss the role of variation in Chloride concentrations during hospital stay and/or during follow-up on final outcomes. The authors can consider and discuss the paper from Bellino MC et al. Eur J Intern Med. 2021 Feb;84:32-37.

Author Response

Response to Reviewers

We wish to express our appreciation to the reviewers and the editor to improve the presentation and understanding of our data. We believe that the comments and suggestions were valuable and we feel the comments have helped us significantly improve the paper. We have considered all of the points raised by the reviewers and provide point-by-point replies to the issues and points the reviewers addressed. As per request of a reviewer, Tables (Table 1, 2 and 3) have been added in the revised version of our manuscript. In addition, a new section titled ‘Clinical perspective’ has been included.

# Reviewer 2

We appreciate the reviewer for careful reading and helpful comments. Respectfully, we have revised the manuscript according to the comments of the reviewer. The comments were valuable and we feel the comments have helped us significantly improve the paper.

Comment 1:

I think that authors should include clinical studies. The inclusion of population studies would increase the appeal of the text and the scientific background.

Response:

 I agree with the reviewer’s comment that the inclusion of population studies would increase the appeal of the text and the scientific background of our paper. In accordance with the reviewer’s suggestion, we have included a new section titled ‘Clinical perspectives’ which mentioned the association of genetic polymorphisms of TMEM16A and hypertension in human. We have added the sentences along with supporting references in the text as follows:

Lines 346-352: While many animal studies suggest that the upregulation of TMEM16A and NKCC1 could contribute to the increased vascular contractility and elevated blood pressure in hypertension as mentioned in the preceding sections, there is very little information concerning their possible involvement in the pathogenesis of hypertension in humans. Interestingly, however, two independent population-based studies reported that some genetic variants of TMEM16A were significantly associated with hypertension in humans [57], [109]. Further exploration of the functional impact of the SNP in the TMEM16A coding region could provide a clue to understand the pathophysiological role of TMEM16A in human hypertension.

Ref.

57. Askew Page, H. R.; Dalsgaard, T.; Baldwin, S. N.; Jepps, T. A.; Povstyan, O.; Olesen, S. P.; Greenwood, I. A. TMEM16A Is Implicated in the Regulation of Coronary Flow and Is Altered in Hypertension. Br.   J. Pharmacol. 2019, 176 (11), 1635–1648. https://doi.org/10.1111/BPH.14598.

109. Jin, H.-S.; Jung, D.  Gender-Specific Association of the ANO1 Genetic Variations with Hypertension . Biomed. Sci. Lett. 2015, 21 (3), 144–151. https://doi.org/10.15616/bsl.2015.21.3.144.

Comment 2:

Authors should include tables which gather the main findings from literature. This would increase the readability of the text.

Response:

As per suggested, Tables (Table 1, 2 and 3) summarizing the published data on the different models have now been added.

Comment 3:

The authors should better discuss the role of variation in Chloride concentrations during hospital stay and/or during follow-up on final outcomes. The authors can consider and discuss the paper from Bellino MC et al. Eur J Intern Med. 2021 Feb;84:32-37.

Response:

We appreciate the appropriate comments of the reviewer. As suggested, we have added the sentences along with supporting references in the ‘Clinical perspectives’ section as follows:

Lines 353-362: In addition, it has been reported that there was a positive association between hyperchloremia and in-hospital mortality in hospitalized patients [110].

Moreover, a recent study in patients with chronic kidney disease, hyperchloremia was an independent predictor of hypertension and proteinuria [111]. Taken these observational studies together, it might be possible to speculate that hyperchloremia might lead to blood pressure elevation and hence to poor prognosis. By contrast, in outpatients with hypertension [112] or chronic heart failure [113], hypochloremia was a predictor of mortality [112], [113] but the level of Clwas not associated with the level of blood pressure [112]. Thus, while these findings indicate that serum Clalterations are associated with poor prognosis in patients with elevated cardiovascular risk, the ability of changes in serum Cl concentration to affect blood pressure is not clear. Further studies are needed to clarify the role of serum Clconcentrations on blood pressure regulation and its association with long-term prognosis in patients with elevated cardiovascular risk.

Ref.

  1. Thongprayoon, C.; Cheungpasitporn, W.; Hansrivijit, P.; Thirunavukkarasu, S.; Chewcharat, A.; Medaura, J.; Mao, M. A.; Kashani, K. Association of Serum Chloride Level Alterations with In-Hospital Mortality. Postgrad. Med. J. 2020, 96 (1142), 731–736. https://doi.org/10.1136/POSTGRADMEDJ-2019-137270.
  2. Takahashi, A.; Maeda, K.; Sasaki, K.; Doi, S.; Nakashima, A.; Doi, T.; Masaki, T. Relationships of Hyperchloremia with Hypertension and Proteinuria in Patients with Chronic Kidney Disease. Clin.Exp.Nephrol. 2022, 26 (9). https://doi.org/10.1007/S10157-022-02229-6.
  1. McCallum, L.; Jeemon, P.; Hastie, C. E.; Patel, R. K.; Williamson, C.; Redzuan, A. M.; Dawson, J.; Sloan, W.; Muir, S.; Morrison, D.; McInnes, G. T.; Freel, E. M.; Walters, M.; Dominiczak, A. F.; Sattar, N.; Padmanabhan, S. Serum Chloride Is an Independent Predictor of Mortality in Hypertensive Patients. Hypertens. (Dallas, Tex. 1979) 2013, 62 (5), 836–843. https://doi.org/10.1161/HYPERTENSIONAHA.113.01793.
  2. Bellino, M. C.; Massari, F.; Albanese, M.; Ursi, R.; Angelini, G.; Lisi, F.; Amato, L.; Scicchitano, P.; Guida, P.; Brunetti, N. D.; Di Serio, F.; Ciccone, M. M.; Iacoviello, M. Baseline and Incident Hypochloremia in Chronic Heart Failure Outpatients: Clinical Correlates and Prognostic Role. Eur. J. Intern. Med. 2021, 84, 32–37. https://doi.org/10.1016/J.EJIM.2020.08.021.

Round 2

Reviewer 2 Report

authors well addressed my previous comments. the paper improved very much